# An Updated Review of Cardiovascular Events in Giant Cell Arteritis

**DOI:** 10.3390/jcm11041005

**Published:** 2022-02-15

**Authors:** Hubert de Boysson, Achille Aouba

**Affiliations:** 1Department of Internal Medicine, Caen University Hospital, Avenue de la Côte de Nacre, 14000 Caen, France; aouba-a@chu-caen.fr; 2Caen University-Normandie, 14000 Caen, France

**Keywords:** giant cell arteritis, large vessel vasculitis, cardiovascular event, stroke, myocardial infarction, peripheral artery disease, aortic dilation, aortic dissection

## Abstract

Giant cell arteritis (GCA) is a systemic vasculitis with a direct and indirect increased risk of acute and chronic vascular events, affecting large and medium vessels, and responsible for most of the morbidity and mortality of this disease. We aimed in this review to provide an updated synthesis of knowledge regarding cardiovascular events observed in GCA. By definition, GCA patients are over 50 and often over 70 years old, and subsequently also present age-related cardiovascular risk factors. In addition, the systemic and vascular inflammation as well as glucocorticoids (GC) probably contribute to an accelerated atherosclerosis and to vascular changes leading to arterial stenoses and aortic dilations and/or dissections. GCA-related ischemic complications, especially ophthalmologic events, stroke or myocardial infarcts are mostly observed within the first months after the diagnosis, being mainly linked to the vasculitic process. Conversely, aortic complications, including dilations or dissections, generally occur several months or years after the diagnosis, mainly in patients with large-vessel vasculitis. In these patients, other factors such as atherosclerosis, GC-related endothelial damage and vascular wall remodeling/healing probably contribute to the vascular events. GCA management includes the detection and treatment of these previous and newly induced cardiovascular risk factors. Hence, the use of cardiovascular treatments (e.g., aspirin, anticoagulation, statins, anti-hypertensive treatments) should be evaluated individually. Aortic structural changes require regular morphologic evaluations, especially in patients with previous aortitis. The initial or secondary addition of immunosuppressants, especially tocilizumab, an anti-IL-6 receptor antibody, is discussed in patients with GCA-related cardiovascular complications and, more consensually, to limit GC-mediated comorbidities.

## 1. Introduction

Giant cell arteritis (GCA) is the most frequent systemic vasculitis in patients over 50, affecting medium- and large-sized vessels. The disease presentation is polymorphic and affects the cranial area in >80% of patients [1,2]. The vascular inflammation of the branches deriving from the external carotid explains most cephalic symptoms observed in the disease such as headaches, scalp tenderness, or jaw claudication. Involvement of ophthalmologic arteries explains visual symptoms described in a quarter of patients at GCA diagnosis and the initial severity of the disease, given the high risk of cecity in the absence of rapid treatment.

A large-vessel vasculitis (LVV; i.e., inflammation of the aorta and its branches) is also described in 40–50% of patients with systematic imaging, although it is often non-symptomatic [3,4]. Therefore, recent guidelines advise to specifically search for LVV at GCA diagnosis [5,6,7].

As an inflammatory vascular disease, GCA can be directly and indirectly responsible for cardiovascular (CV) events. Thanks to large-registry studies, it is nowadays well-established that GCA is associated with an increased risk of CV events [8,9,10], increasing the mortality from CV origin [11,12,13]. By definition, GCA patients are over 50 and they are even more frequently between 70 and 80 years old, an age at which usual cardiovascular risk factors and atherosclerosis are also common [14].

Regarding CV risk factors, some registry studies dealing with >5000 GCA patients and >30,000 sex- and age-matched controls indicated that patients with GCA more frequently display CV risk factors, especially dyslipidemia and hypertension [9,15]. They also more frequently present at diagnosis a history of angina pectoris, ischemic heart disease, stroke, peripheral vascular disease and aortic aneurysm than controls [9].

Upon analyzing the occurrence of different vascular diseases during follow-up, GCA patients have an increased risk of myocardial infarction [8,9,10], cerebrovascular events [8,9,10,15], peripheral vascular disease [9,10] and aortic aneurysms [9,16,17,18] than sex- and age-matched controls [9,15].

In this review, we aimed to depict the CV complications described in GCA. In the first part, we focused on the ischemic events observed in the disease. We delineated in the second part all the complications affecting the aorta and its main branches. Finally, we discussed the therapeutic management of a GCA patient exhibiting a cardiovascular event.

## 2. Ischemic Events in GCA

### 2.1. Implicated Mechanisms

Ischemic events occurring in GCA might be linked to three distinct or intricated mechanisms.

The first mechanism corresponds to the vascular wall inflammation that leads to a narrowing of the vascular lumen and downstream tissue ischemia. The pathophysiology of vascular damage in GCA is to date well understood. Briefly, following a trigger whose origin is still unknown, dendritic cells in the adventitia of arteries are activated and result in the production of chemokines that attract lymphocytes within the vascular wall. A cytokine-mediated activation, proliferation and polarization of T lymphocytes into TH1 and TH17 cells lead to the recruitment of macrophages and T CD8 lymphocytes. In addition, some growth factors such as vascular endothelial growth factor (VEGF) and platelet-derived growth factor (PDGF) increase the proliferation of vascular smooth muscle cells. Altogether, the different recruited and activated cells are responsible for vascular damage and result in an intimal hyperplasia that can progressively obstruct the vascular lumen [19,20,21].

The second mechanism regards the possible accelerated atherosclerosis in patients with GCA, also documented in systemic lupus erythematosus, rheumatoid arthritis and in some systemic vasculitides [22,23,24,25]. Endothelial dysfunction, which is a cornerstone of atherogenesis [26], has been poorly studied in GCA. Measuring the flow-mediated vasodilatation by brachial ultrasonography, which is a marker of endothelial function, two studies provided paradoxical results. One showed an initial impaired flow-mediated vasodilatation that improved with GC [27] supporting the presence of an endothelial dysfunction in GCA patients before treatment. However, these results were not replicated in another study [28].

Another ultrasonography study did not show any increase in the rate of atheromatous carotid plaques during the follow-up of GCA patients when compared to controls [29]. Thus, taken together, no specific study has shown accelerated atherosclerosis in GCA patients, although there is much evidence that systemic inflammation provides a proatherogenic effect by inducing dyslipidemia, insulin resistance, hypercoagulation, endothelial dysfunction and oxidative stress [22].

The third mechanism corresponds to the various direct and indirect impacts of GC on atherosclerosis formation. Glucocorticoid excess, as observed in Cushing’s syndrome, is associated with an increased risk of CV events [30]. A recent study of 87,794 patients with immune mediated diseases, including 25,581 GCA patients +/− with polymyalgia rheumatica, confirmed that GC use, even with daily doses <5 mg of prednisone equivalent, was associated with an increased risk of CV disease [31]. However, the implication of GC in atherogenesis is complex, since the effects of GC were shown to be both athero-protective and pro-atherogenic. On one hand, some preclinical studies showed that GC exposure may reduce the recruitment of monocytes and macrophages within the atherosclerotic lesions and reduce the cholesterol accumulation in macrophages. It also results in a reduction of macrophage-induced pro-inflammatory cytokine release limiting the vascular remodeling [32]. On the other hand, GC are also implied in pro-atherogenic mechanisms. They lead to a direct endothelial cell dysfunction by promoting the production of vasoconstrictors acting on the smooth muscle cells. In addition, some other pre-clinical experimental models suggest an increased accumulation of macrophages in atherosclerotic lesions, with an increased cholesterol esterification [32]. Altogether, long term GC exposure is associated with increased atherosclerosis. However, the exact impact of GC and the mechanisms implied in atherosclerosis formation is more complex, since GC may act at different cellular levels resulting in opposite results, either beneficial or detrimental to the development of atherosclerotic plaques.

### 2.2. Distinction of Vascular Inflammation and Atherosclerosis in Ischemic Events

In patients with GCA, the occurrence of an ischemic raises the question of the implicated vascular mechanism. Disregarding embolic causes, atherosclerosis and vascular inflammation are the two other main mechanisms that can lead to CV events.

In most cases, the imputability of the vasculitis in the CV event relies on the association of indirect evidence. Since imaging rarely demonstrates a circumferential and homogeneous thickening of the vessel implied in the downstream ischemia, vasculitis is mainly suspected in patients with active disease, i.e., with increased acute phase reactants. Moreover, vasculitic-related ischemic events (mainly stroke and myocardial infarctions) often occur at an early stage of the disease when systemic and vascular inflammation is still uncontrolled. Conversely, ischemic events occurring in patients with inactive or untreated disease are more likely linked to atherosclerotic mechanisms, especially when many CV risk factors coexist.

Some vascular territories are more likely affected in vasculitis-related CV events, which can help determine the involved mechanism. As an example, the vertebrobasilar arteries or the left main coronary artery are more frequently affected in patients with GCA-related stroke or myocardial infarction, respectively, when compared to matched controls [33,34].

### 2.3. Retinal Ischemia

Visual symptoms (blurred vision, unilateral or bilateral cecity, diplopia) are frequent at GCA diagnosis and are reported in 10% to 28% of patients [35,36]. These symptoms are particularly feared because possibly related to the occlusion of the ophthalmologic arteries and their branches with a risk of definite blindness or important visual damage in two thirds of these patients. Typically, GCA-related involvements of the posterior ciliary arteries or the central retinal artery lead, respectively, to acute anterior ischemic optic neuropathy (AION) or central retinal artery occlusion (CRAO). However, once treatment is started, the control of the systemic and vascular inflammation drastically reduces the risk of a new ophthalmologic ischemic event. Even during GCA relapses, that affect almost half of patients, AION and CRAO remain exceptional [37,38].

Of note, CRAO is frequently associated with atheromatous lesions and in the absence of increased acute phase reactants, other causes than GCA should be searched. Several studies suggest a link between inflammatory ophthalmologic involvement and atherosclerosis, since visual losses found in GCA were more likely to occur in older patients with past histories of stroke and peripheral arterial disease [36,39,40]. 

### 2.4. Stroke

After retinal ischemia, cerebrovascular events are the second leading CV event in GCA, with a higher risk of occurrence than in the general population [8,10,15]. A systematic review and meta-analysis established a risk ratio of cerebrovascular events of 1.40 [95% CI: 1.27—1.56] when compared to non-GCA control patients [41].

Important discrepancies exist among studies dealing with stroke in GCA patients. Some only focused on stroke occurring within the first weeks following GCA diagnosis assuming that this design allows the capture of stroke directly related to the vasculitic process. Others analyzed stroke occurring at any time in GCA patients and thus probably included non-GCA-related stroke. Finally, some included transient ischemic attacks or retinal ischemia in the analyzed cerebrovascular events [42]. Altogether, comparisons of these different studies are difficult.

Many authors agreed that GCA-related strokes mostly occur within the first weeks following the onset of the disease and are mostly of ischemic nature. In this restricted post-diagnosis timeframe, the incidence of GCA-related stroke ranged in different retrospective studies from 2.7% to 7.4% of GCA patients [33,39,43,44,45,46]. 

Some common findings were observed regarding GCA-related stroke in these studies. A predominance of male gender was observed in several studies [33,39,45]. The vertebrobasilar territory was affected in 50 to 100% of GCA-related strokes while it was only involved in one fifth of strokes observed in the general population [47,48]. Ultrasonographic studies frequently showed an inflammatory vascular thickening and stenoses/occlusions of the vertebral arteries in patients with recent GCA diagnosis and posterior stroke [49,50], underlining the role of the vasculitic process in the cerebrovascular event. However, in a prospective study in which every consecutive patient with vertebrobasilar stroke had an ultrasonographic examination, only 2 out of 65 patients (3.1%) had vascular inflammatory findings—i.e., the halo sign—suggestive of GCA [51]. These two patients were older and exhibited increased acute phase reactants. Intracerebral vasculitis directly responsible for GCA-related stroke appears very rare and is still a matter of debate considering that intracranial vessels contain no or little internal elastic lamina which constitutes the supposed anatomic substratum targeted by the pathophysiologic inflammatory process specifically involved in GCA [52]. In patients with supratentorial stroke, GCA is often suggested by the presence of specific associated cranial symptoms and the presence of increased acute phase reactants. The halo sign on the arteries from the carotid territory is rare and, when GCA is suspected, ultrasonographic examination should include the superficial temporal arteries that are involved in the majority of cases [33,47].

Some studies identified some predictive factors for stroke in GCA patients: male gender [39,45], the presence of GCA-related visual symptoms [33,39,43,45], a history of hypertension [39,45] and smoking [39], or the absence of anemia that often reflects a lower systemic inflammation [33,39,43,44,45].

Cerebrovascular events remain one of the major causes of mortality in patients with GCA. In a multicenter retrospective study, 28% of patients with GCA-related stroke died in the following days or weeks [33].

### 2.5. Acute Coronary Syndrome

Few studies are specifically dedicated to the analysis of the risk of acute coronary events in GCA patients. Two population-based studies did not demonstrate an increased risk of acute coronary syndrome in patients with GCA [53,54], whereas others suggested an increase [8,9,10,55].

The incidence rate of ischemic heart disease in GCA patients ranged between 5 and 9% [34,53,54]. Only one cohort study tried to differentiate acute coronary syndrome caused by GCA from those associated with atherothrombotic coronary artery disease [34]. In this study, myocardial infarction with coronary angiography demonstrating obstructive lesions in the absence of a GCA flare was considered more likely caused by atherothrombotic disease. Conversely, myocardial infarction occurring in the 3 months preceding or following a GCA flare was more likely considered linked to the vasculitis. Interestingly, 80% (4/5) of coronary angiographies from patients with GCA-related myocardial infarctions showed no evidence of plaque rupture but conditions triggering an imbalance between demand and supply of myocardial oxygen. Based on this distinction, this study identified a lower rate of acute coronary syndrome directly linked to GCA (6/251 = 2.4%) [34]. When comparing patients from both groups, patients with GCA-related myocardial infarctions were younger (median of 79 versus 86 years in the atherothrombotic group). Although the rates of CV risk factors were similar in both groups, no patient with GCA-related myocardial infarction had a past history of CV event. The GCA characteristics were not different in the two groups, but patients whose myocardial infarction was not related to GCA showed higher cumulative GC doses.

Vasculitis of coronary arteries is exceptional [34,56]. Myocardial infarction in the setting of a GCA flare could be due to systemic inflammation that negatively affects cardiac function and may lead to myocardial ischemia when oxygen supply is not sufficient or when cardiac preload decreases because of inflammation-mediated vasodilation [57]. 

The study of Greigert et al. also strongly suggested that most myocardial infarctions are not linked to GCA, i.e., are not associated with systemic inflammation; indeed, these ischemic events occurred several years post-diagnosis in patients exhibiting a long-term remission of the vasculitis [57]. Finally, the survival of patients with GCA-related myocardial infarctions was not more limited than non-GCA control patients (one-year survival rate of 83%). Conversely, GCA patients with atherothrombotic myocardial infarctions had shorter survival than non-GCA controls (one-year survival rate of 50% versus 91% in controls, *p* = 0.004) [57].

### 2.6. Peripheral Artery Disease

As for stroke and myocardial infarctions, some registry studies indicated a higher risk of limb peripheral vascular disease in GCA patients [9,10]. However, few data exist in the literature to determine the frequency of peripheral artery disease in GCA patients. Two main difficulties explain why such information is lacking. First, peripheral arteries, especially ilio-femoral arteries and downstream branches, are more frequently affected by atherosclerosis in the elderly population with GCA, obfuscating the distinction between a vascular claudication linked to an inflammation of peripheral arteries and a claudication secondary to atheromatous lesions. Even with the help of imaging, the distinction of both disorders remains difficult. Second, unlike stroke and myocardial infarction which correspond to acute events that are easily captured, peripheral artery disease includes different components, ranging from a chronic vascular claudication evolving into a progressive then subacute ischemia, to an acute limb ischemia. Except for certain studies dealing only with vasculitis-related limb ischemia, all others analyzing peripheral artery disease in GCA included both chronic and acute events.

In the absence of treatment, inflammation of the subclavian, supra-aortic, or ilio-femoral arteries can slowly evolve toward vascular stenosis or occlusion, that can subsequently be responsible for a clinical ischemic complication.

Before the era of imaging in GCA patients, lower extremity vasculitis in GCA patients was rarely reported. As an example, a systematic review of medical records from 6212 patients evaluated for GCA between 1983 and 2007 at the Mayo Clinic, Rochester, identified only 19 patients with lower extremity vasculitis, diagnosed through lower limb claudication and elevated inflammatory markers [58]. In another population-based study of 168 patients followed between 1950 and 1999, 13% of them developed clinical large-artery stenosis, including 9% with incident cervical artery stenosis and 4% with subclavian/axillary/brachial artery stenosis. Only one patient (0.6%) had incident ilio-femoral artery stenosis [59]. On the other hand, in a retrospective sonographic study of 60 patients followed in a Vascular Medicine unit, Czihal et al. observed vasculitis on femoropopliteal arteries in 32 (53%), with accompanying symptoms in only 44% (14/32; i.e., 14/50 = 23% of the entire cohort). Interestingly, three fourths of patients with femoropopliteal involvement presented concomitant vasculitis of the upper limbs [60]. Although the patients from this study were selected in a Vascular Medicine department, it shows that systematic imaging of the lower limb arteries in GCA patients identified a significant rate of non-symptomatic involvement.

Taken together, it is impossible to determine the exact prevalence of peripheral artery disease in GCA and, *a fortiori,* those leading to ischemic complications. Analysis of case series reporting GCA-related peripheral vessel involvement showed some common findings, especially a predominance in younger (<70 years) females [61,62,63]. The diagnosis is often delayed [63], especially when cranial symptoms lack. Increased acute phase reactants often guide the clinician toward GCA diagnosis. Notably, one quarter to half of all patients with symptomatic limb involvement require acute vascular surgery for ischemic signs [61,63].

### 2.7. Other Ischemic Complications

Considering the possible involvement of branches from the aorta and their subdivisions, some reports describe vasculitis of visceral arteries with ischemic complications such as medullary or mesenteric infarctions [64,65]. These cases remain exceptional and the diagnosis of GCA often relies on the presence of cranial symptoms, increased inflammatory parameters or concomitant LVV.

## 3. Aortic Complications in GCA

To define the CV events that can affect the aorta during GCA, the generic headings “aortic complications” or “aortic events” are often used. These terms often define the occurrence of any change of aorta morphology, especially dilation/aneurysm or dissection. Although the definitions of “aneurysm” and “dilation” are different, the words used in different studies often define the change and increase of aorta caliber. To remain homogeneous in the following chapter, we will only use the word “dilation”. Moreover, we included under the term “aortic complications” the occurrence of any aorta dilation or aortic dissection during the GCA course.

The exact incidence of aortic dilation in GCA, at diagnosis or during follow-up, is unknown since the recommendation to perform imaging at GCA diagnosis is recent [6], and many patients do not have any control of the aorta morphology during follow-up. Aortic dissection in the setting of GCA is very rare and its incidence was previously estimated at 1.9 per 1000 Person Years [66].

Before the era of large vessel imaging, some cohort studies suggested the possible involvement of large vessels and the subsequent risk of aortic dissection in GCA. In 1975, Klein et al. detected a clinical large artery involvement in 34/248 (14%) GCA patients and 3 of them died from an aortic rupture; autopsy revealed an active giant cell vasculitis in all three [67]. In another population-based study from 1995, Evans et al. suggested that GCA patients have a 17-fold higher risk of developing an aortic dilation when compared to the general population [16]. More recently, the analysis of a UK registry of 6999 patients with GCA compared to 41,994 sex- and aged-matched controls, finally found, after adjustment for cardiovascular risk factors, a twofold increased risk of aortic dilation [17]. Thus, the higher risk of aortic dilation in GCA patients is unanimously admitted today. Other cohort studies confirmed that patients with aortic dilation have an increased mortality, especially secondary to aortic dissections [11,12,13,68,69].

In the early 2000s, Blockmans et al. first described the routine use of positrons emission tomography with fluorodeoxyglucose (^18^FDG-PET/CT) in GCA patients. This team demonstrated the high sensitivity of this imaging tool for the detection of LVV [70,71]. They also were the first to suggest that persisting aortic hypermetabolism might later favor the occurrence of aortic dilation [72]. Since then, many other cohort studies confirmed that GCA patients with LVV have a higher risk of developing an aortic dilation, mostly on the involved arterial segments, versus those without LVV [73,74,75,76]. Moreover, while aortic complications are often discovered between 5 and 10 years after GCA diagnosis [18,68], patients with LVV might develop aortic dilation or dissection earlier than those without [74,77].

Following these works, some authors demonstrated that LVV was a prognostic factor in GCA patients, since it may favor disease relapse [78,79,80] and development of any kind of CV event during the follow-up [74,77,81], especially if aortitis was symptomatic [82].

Male sex [83] or female sex [18], hypertension [66,83], hyperlipidaemia [59] or coronary disease [59] has been also identified as predictive factors for aortic dilation in patients with LVV.

The mechanisms that underlie the higher rate of aortic events in patients with LVV are partially understood. Since some observations reported active vasculitis on aorta histology from patients who experienced an aortic dissection [67,77,81], we can hypothesize that the inflammation within the aorta weakens the vessel wall and favors the subsequent occurrence of dilation or dissection. In addition, aortic dissections in the setting of LVV often occur on a previously inflamed segment [77]. Interestingly, even in patients with clinical and biological remission, a subclinical inflammation may persist within the aorta wall, although not detectable through systemic acute phase reactants. Beside the inflammatory hypothesis, the impact of healing or remodeling processes on the aorta architecture is unknown. Moreover, usual CV risk factors such as hypertension may add to the risk of dilation.

Altogether, based on the previous studies showing the frequent development of a dilation mostly on the thoracic section of the aorta in patients with LVV [73,74,75,76], a regular monitoring of aorta caliber is required. Echocardiography may be a useful, cost-effective and non-irradiating tool to monitor the thoracic aorta caliber and detect any dilation. An annual echocardiography, associated with the overall assessment of usual CV risk factors in this elderly population, could be suggested. However, given the absence of specific guidelines regarding the best frequency and modality of aorta monitoring, further research is required to confirm the usefulness of this practice in GCA.

## 4. Therapeutic Approaches

### 4.1. Specific Treatment of GCA

In the last few years, the European League Against Rheumatism (2018; EULAR [7]) followed by the American College of Rheumatology (2021; ACR [84]) updated their recommendations for the treatment of GCA as did the French (2016 and 2020) [5,85] and British (2021) [86] specific societies.

An update of previous recommendations was required for two main reasons. First, the studies of the last 15 years emphasized the high prevalence of LVV in GCA patients with a possible poorer prognosis in patients with LVV. Subsequently, recommendations for the use of imaging and the specific treatment of LVV were required. Secondly, in 2017, the Giacta trial led to the authorized use of tocilizumab (TCZ), an anti-IL-6 receptor monoclonal antibody, in the treatment of GCA [87]. This trial demonstrated the beneficial effect of TCZ on GC sparing, since patients who received this treatment cumulated lower GC doses 3 years after inclusion [87,88]. Similarly, in the same time period, many studies showed that the modern challenge of GCA care was to reduce the exposure to GC, given the many adverse events of this treatment [89,90]. 

American and European recommendations were thus updated to include the different radio-clinical patterns of GCA and define the place of TCZ in the therapeutic strategies. Some disparities exist in both recommendations, especially regarding GC management and the use of TCZ.

ACR and EULAR agreed that GC remain mandatory in the treatment of GCA. In patients without ischemic event (no visual involvement nor stroke), high-dose oral GC are recommended (starting at 40 to 60 mg/day as recommended by EULAR [7]; no specification of dose in the ACR guidelines [84]). In patients with ischemic event, both recommendations conditionally recommend the use of IV pulses of methylprednisolone (0.25 to 1 g for up to 3 days in the EULAR guidelines). While EULAR recommends tapering GC to target 15–20 mg/day within 2–3 months and ≤5 mg/day after one year [7], ACR do not advise specific dosing and duration, assuming these parameters are variable depending on patients’ characteristics and comorbidities. More importantly, EULAR advises the adjunction of an immunosuppressant (TCZ at first, methotrexate as an alternative) only in selected patients with refractory or relapsing disease, or with GC-related complications, whereas ACR conditionally recommends the use of oral GC associated with TCZ over GC alone in newly diagnosed GCA assuming the importance of sparing GC [7,84]. EULAR does not recommend adjunctive therapy in all GCA patients given the substantial number of patients able to taper GC to under 5 mg/day after one year [7].

LVV should be treated with GC combined with an immunosuppressant (TCZ or methotrexate) according to ACR, while EULAR only advises adding an immunosuppressant in patients with relapsing disease or with GC-related comorbidities [7,84]. 

Relapse management also differs between both recommendations. The EULAR taskforce recommends treating relapse with a dose escalade of GC, up to doses initially prescribed at new-onset disease (i.e., 40–60 mg/day) in case of major relapse (either with signs of ischemia or progressive vascular inflammation). Otherwise, GC dose should be increased to at least the last effective dose. In case of multiple relapses, or in patients with poor GC tolerance, an adjunctive immunosuppressant should be discussed [7]. According to ACR, the first relapse should be systematically treated with the addition of an immunosuppressant [84].

In the published literature, more data exist on the use of TCZ in comparison to methotrexate for GC-sparing strategies, explaining the predominant choice of TCZ rather than methotrexate [87,91]. However, methotrexate is probably a good cost-effective alternative in patients with TCZ contraindications or inefficacy [92]. Only few data exist on the effectiveness of TCZ or methotrexate on LVV [91,93,94]. Of note, a recent observational study showed a good tolerance and efficacy profile of TCZ in patients over 80 years [95].

### 4.2. Antiplatelet, Anticoagulation and Statins

The prescription of antiplatelet, anticoagulation or statins should be individually discussed in each patient according to CV risk factors and comorbidities [7,84].

Regarding the potential benefit of aspirin in the prevention of GCA-related CV complications, studies are controversial. Some showed a benefit [96,97], whereas others did not [44,98]. Finally, a meta-analysis did not indicate a beneficial effect in patients with previous antiplatelet prescription before GCA, but a slight favorable outcome in patients who received GC + aspirin at GCA onset [99]. ACR and EULAR do not recommend the routine prescription of aspirin but conditionally recommend its prescription in patients with critical or flow-limiting involvement of carotid or vertebral arteries (ACR) or in patients with vascular ischemic complications (EULAR) [7,84].

Regarding statins, in the absence of a separate indication recommending their use, ACR recommends not using them systematically in the treatment of GCA [84].

## 5. Conclusions

In this review, we aimed to underline how giant cell arteritis is a systemic vascular disease, with a direct and indirect increased risk of vascular events that remain the major cause of morbidity and mortality in this disease. On one hand, cardiovascular complications in GCA are due to the vasculitic process that can participate in the vascular narrowing or weakness that lead, respectively, to ischemic events and aortic dilation/dissection. On the other hand, atherothrombotic events are also more prevalent in GCA, probably favored by inflammation-mediated accelerated atherosclerosis, direct GC-induced structural and functional damage of the endothelium, and the frequent cumulative effect of classical cardiovascular risk factors in this predominantly elderly population. GCA-related ischemic complications, especially ophthalmologic events, stroke or myocardial infarction are mostly observed within the first month post-diagnosis and are mainly linked to the vasculitic process. Conversely, aortic complications (i.e., aorta dilation or dissection) generally occur years after the diagnosis, mainly in patients with previous large-vessel vasculitis (i.e., inflammation of the aorta and its main branches).

GCA treatment implies an individual evaluation of the cardiovascular risk. The vasculitis should be treated with GC and the use of immunosuppressants, especially tocilizumab, should frequently be discussed in patients with GCA-related cardiovascular complications and, more consensually, to limit GC-mediated comorbidities. In the near future, other targeted molecules will probably be added to the therapeutic arsenal of this disease, as suggested by the high number of ongoing clinical trials [abatacept (NCT04474847), anakinra (NCT02902731), guselkumab (NCT04633447), mavrilimumab (NCT038227018), secukinumab (NCT04930094), upadacitinib (NCT03725202), or ustekinumab (NCT03711448)]. The therapeutic challenge is nowadays to rapidly control the systemic and vascular inflammation, decreasing *de facto* the risk of ischemic events, and reducing the long-lasting harmful effect of glucocorticoids and vascular remodeling that favor late cardiovascular and aortic events. Personalized treatment is the next step in GCA.

Further works are awaited to better determine and act on the mechanisms involved in cardiovascular events in GCA patients, especially regarding the endothelial dysfunction that leads to the intimal hyperplasia or vascular remodeling. Besides anti-inflammatory or immunosuppressive treatments, future therapeutic approaches developed in atherosclerosis and aiming to limit plaque formation or to reduce the involved inflammatory processes may be beneficial in GCA.

## Data Availability

Not applicable.

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
