# Peer review of "An Updated Review of Cardiovascular Events in Giant Cell Arteritis"

_jcm, 2022, doi:10.3390/jcm11041005_

Round 1

Reviewer 1 Report

The topic of this review is relevant for physicians caring for patients with GCA

general comment

  1. there is a rendundancy in text within the different chapters. mechanisms of CV events are discussed in several chapters. 
  2. Manifestations of CV events in GCA are repeated e.g. line 67 Some CV events, especially cerebrovascular or retinal ischemia, mainly occur at an early stage of the disease and are probably the consequences of the inflamed narrowing of the affected vessels. ... line 90 The first mechanism corresponds to the vascular wall inflammation that 
    leads to a narrowing of the vascular lumen and downstream tissue ischemia. Typically, this process is involved early in the disease, when systemic and vascular inflammation are not yet controlled by treatments. Typically, ophthalmologic involvements proceed ....and again in  chapter 3. another example for redundnacy of  line 78-79 and 97
  3. Citation is often missing especially chapter 3 without any citation 3.1. statements without citation
  4.  pasted text from another review: line 86 By definition, GCA patients are over 50 and they are even more frequently between 70 and 80 years old, an age at which usual cardiovascular risk factors and atherosclerosis are also common. the original: 

    “Importantly, by definition GCA only develops in patients after the age of 50 years, and most typically presents in those ages 75 to 85 years--an age when atherosclerosis is common” (https://www.sciencedirect.com/science/article/pii/S0021915021001532?via%3Dihub)

    the source is only cited in another context
  5. original literature is incorrectly cited line 61 Vascular disease (myocardial infarction, ischemic heart disease, cerebrovascular events, peripheral vascular disease and aortic aneurysms) was also more prevalent at diagnosis in GCA patients than in sex- and age-matched controls the cited reference: Patients with GCA were more likely to have a history of vascular diseases and other comorbidities except myocardial infarction” (https://pubmed.ncbi.nlm.nih.gov/28077689/)
  6. In the third chapter  aortal dilataion is not a mechanism of ischemic event
  7. The abstract does not clearly state the aims of the review
  8. The limitations of distintcion of vasculitic ischemic events and arteriosclerotic events should be discussed and separately listed 
  9. pathophysiology are hypothetical and should be in depth discussed in a separate chapter (disease related, arteriosclerotic and the effects of Prednisone) maybe a figure would be illustrative
  10. tables would be helpful 
  11. Finally a research agenda would be useful

Author Response

RESPONSE TO REVIEWER 1

  1. There is a redundancy in text within the different chapters. mechanisms of CV events are discussed in several chapters. 

We modified our manuscript in order to reduce redundancies. In accordance with the following queries, we added a paragraph on the mechanisms implied in ischemic events. We thus deleted in the other chapters all information regarding these different mechanisms.

  1. Manifestations of CV events in GCA are repeated e.g. line 67 Some CV events, especially cerebrovascular or retinal ischemia, mainly occur at an early stage of the disease and are probably the consequences of the inflamed narrowing of the affected vessels.

      ... line 90 The first mechanism corresponds to the vascular wall inflammation that
      leads to a narrowing of the vascular lumen and downstream tissue ischemia.            Typically, this process is involved early in the disease, when systemic and vascular   inflammation are not yet controlled by treatments. Typically, ophthalmologic          involvements proceed ....and again in  chapter 3. another example for redundnacy       of  line 78-79 and 97

Thank you for this careful analysis.

We deleted redundant sentences in our manuscript, and we deleted the chapter " Global characteristics of GCA patients and cardiovascular events”.

  1. Citation is often missing especially chapter 3 without any citation 3.1. statements without citation

As suggested in queries 8 and 9, chapter 3 was deeply modified (and is now replaced by chapter 2). The long initial introduction (where references were indeed missing) was deleted and replaced by two new paragraphs (2.1 and 2.2).

  1.  pasted text from another review: line 86 By definition, GCA patients are over 50 and they are even more frequently between 70 and 80 years old, an age at which usual cardiovascular risk factors and atherosclerosis are also common. the original: 

“Importantly, by definition GCA only develops in patients after the age of 50 years, and most typically presents in those ages 75 to 85 years--an age when atherosclerosis is common” (https://www.sciencedirect.com/science/article/pii/S0021915021001532?via%3Dihub). The source is only cited in another context

We added the reference from Clifford et al. page 5 at the end of the first paragraph.

  1. Original literature is incorrectly cited line 61 Vascular disease (myocardial infarction, ischemic heart disease, cerebrovascular events, peripheral vascular disease and aortic aneurysms) was also more prevalent at diagnosis in GCA patients than in sex- and age-matched controls the cited reference: Patients with GCA were more likely to have a history of vascular diseases and other comorbidities except myocardial infarction”

      (https://pubmed.ncbi.nlm.nih.gov/28077689/)

Since this paragraph of the introduction also showed some redundancies, we rewrote it page 4 and 5, and corrected the citation:

" As an inflammatory vascular disease, GCA can be directly and indirectly responsible for cardiovascular (CV) events. Thanks to large-registry studies, it is nowadays well-established that GCA is associated with an increased risk of CV events [8-10], increasing the mortality from CV origin [11-13]. By definition, GCA patients are over 50 and they are even more frequently between 70 and 80 years old, an age at which usual cardiovascular risk factors and atherosclerosis are also common [14].

Regarding CV risk factors, some registry studies dealing with >5.000 GCA patients and >30.000 sex- and age-matched controls indicated that patients with GCA more frequently display CV risk factors, especially dyslipidemia and hypertension [9, 15]. They also more frequently present at diagnosis a history of angina pectoris, ischemic heart disease, stroke, peripheral vascular disease and aortic aneurysm than controls [9].

Upon analyzing the occurrence of different vascular diseases during follow-up, GCA patients have an increased risk of myocardial infarction [8-10], cerebrovascular events [8-10, 15], peripheral vascular disease [9, 10] and aortic aneurysms [9, 16-18] than sex- and age-matched controls [9, 15]."

  1. In the third chapter, aortal dilatation is not a mechanism of ischemic event.

This was deleted.

  1. The abstract does not clearly state the aims of the review

We added in the abstract "We aimed in this review to provide an updated synthesis of knowledge regarding cardiovascular events observed in GCA."

  1. The limitations of distinction of vasculitic ischemic events and arteriosclerotic events should be discussed and separately listed 

We agree that the appreciation of the nature of an ischemic event is difficult in the setting of a vasculitis. Inflammatory vascular narrowing and atherosclerosis are the two main non-exclusive causes that can lead to an ischemic event. In the Chapter on "ischemic events in GCA", after the paragraph "2.1 Implicated mechanisms" (see the following query), we added the following paragraph page 7 "2.2 Distinction of vascular inflammation and atherosclerosis in ischemic events".

However, in order to limit redundancies with the characteristics of each ischemic event (stroke, myocardial infarction, lower limb event) that are detailed in their respective specific paragraphs, we only described the global characteristics that help distinguish vascular inflammation and atherosclerosis.

2.2 Distinction of vascular inflammation and atherosclerosis in ischemic events

In patients with GCA, the occurrence of an ischemic raises the question of the implicated vascular mechanism. Disregarding embolic causes, atherosclerosis and vascular inflammation are the two other main mechanisms that can lead to CV events.

In most cases, the imputability of the vasculitis in the CV event relies on the association of indirect evidence. Since imaging rarely demonstrates a circumferential and homogeneous thickening of the vessel implied in the downstream ischemia, vasculitis is mainly suspected in patients with active disease, i.e. with increased acute phase reactants. Moreover, vasculitic-related ischemic events (mainly stroke and myocardial infarctions) often occur at an early stage of the disease when systemic and vascular inflammation is still uncontrolled. Conversely, ischemic events occurring in patients with inactive or untreated disease are more likely linked to atherosclerotic mechanisms, especially when many CV risk factors coexist.

Some vascular territories are more likely affected in vasculitis-related CV events, which can help determine the involved mechanism. As an example, the vertebrobasilar arteries or the left main coronary artery are more frequently affected in patients with GCA-related stroke or myocardial infarction, respectively, when compared to matched controls [33, 34].

  1. Pathophysiology are hypothetical and should be in depth discussed in a separate chapter (disease related, arteriosclerotic and the effects of Prednisone) maybe a figure would be illustrative

Thank you for this comment. In the chapter on "ischemic events in GCA", we added the following paragraph "Implicated mechanisms" page 5.

Given the complexity and uncertainty of the mechanisms, and the multiple cellular and molecular pathways involved, we fear that a Figure could risk being rather confusing.   

2.1 Implicated mechanisms

Ischemic events occurring in GCA might be linked to three distinct or intricated mechanisms.

The first mechanism corresponds to the vascular wall inflammation that leads to a narrowing of the vascular lumen and downstream tissue ischemia. The pathophysiology of vascular damage in GCA is to date well understood. Briefly, following a trigger whose origin is still unknown, dendritic cells in the adventitia of arteries are activated and result in the production of chemokines that attract lymphocytes within the vascular wall. A cytokine-mediated activation, proliferation and polarization of T lymphocytes into TH1 and TH17 cells lead to the recruitment of macrophages and T CD8 lymphocytes. In addition, some growth factors such as vascular endothelial growth factor (VEGF) and platelet-derived growth factor (PDGF) increase the proliferation of vascular smooth muscle cells. Altogether, the different recruited and activated cells are responsible for vascular damage and result in an intimal hyperplasia that can progressively obstruct the vascular lumen [19-21].

The second mechanism regards the possible accelerated atherosclerosis in patients with GCA, also documented in systemic lupus erythematosus, rheumatoid arthritis and in some systemic vasculitides [22-25]. Endothelial dysfunction, which is a cornerstone of atherogenesis [26], has been poorly studied in GCA. Measuring the flow-mediated vasodilatation by brachial ultrasonography, which is a marker of endothelial function, two studies provided paradoxical results. One showed an initial impaired flow-mediated vasodilatation that improved with GC [27] supporting the presence of an endothelial dysfunction in GCA patients before treatment. However, these results were not replicated in another study [28].

Another ultrasonography study did not show any increase in the rate of atheromatous carotid plaques during the follow-up of GCA patients when compared to controls [29]. Thus, taken together, no specific study has shown accelerated atherosclerosis in GCA patients, although there is much evidence that systemic inflammation provides a proatherogenic effect by inducing dyslipidemia, insulin resistance, hypercoagulation, endothelial dysfunction and oxidative stress [22]. 

The third mechanism corresponds to the various direct and indirect impacts of GC on atherosclerosis formation. Glucocorticoid excess, as observed in Cushing's syndrome, is associated with an increased risk of CV events [30]. A recent study of 87,794 patients with immune mediated diseases, including 25,581 GCA patients +/- with polymyalgia rheumatica, confirmed that GC use, even with daily doses <5mg of prednisone equivalent, was associated with an increased risk of CV disease [31]. However, the implication of GC in atherogenesis is complex, since the effects of GC were shown to be both athero-protective and pro-atherogenic. On one hand, some preclinical studies showed that GC exposure may reduce the recruitment of monocytes and macrophages within the atherosclerotic lesions and reduce the cholesterol accumulation in macrophages. It also results in a reduction of macrophage-induced pro-inflammatory cytokine release limiting the vascular remodeling [32]. On the other hand, GC are also implied in pro-atherogenic mechanisms. They lead to a direct endothelial cell dysfunction by promoting the production of vasoconstrictors acting on the smooth muscle cells. In addition, some other pre-clinical experimental models suggest an increased accumulation of macrophages in atherosclerotic lesions, with an increased cholesterol esterification [32]. Altogether, long term GC exposure is associated with increased atherosclerosis. However, the exact impact of GC and the mechanisms implied in atherosclerosis formation is more complex, since GC may act at different cellular levels resulting in opposite results, either beneficial or detrimental to the development of atherosclerotic plaques.

  1. Tables would be helpful 

We agree that tables can be helpful in providing an overview of a topic.

We initially planned to summarize all population-based studies analyzing cardiovascular events in GCA in a Table format. However, we did not find that a Table was highly contributive or increased the clarity of our presentation, since all studies found similar results and important discrepancies existed between the sample sizes, timeframes, definition of events, etc. These different studies are described in the text.

Regarding other potential Tables, we found them too simplistic (description of different CV events , risk factors, or treatment used, etc.)

  1. Finally, a research agenda would be useful

Many works have already been done regarding CV events in GCA. However, some questions remain unanswered and open the path to further studies.

We did not find studies specifically dedicated to CV events in GCA in the ClinicalTrial database. One study found through ClinicalTrial (NCT03841734) aimed to analyze the early benefits of endothelin inhibitors in GCA patients with sudden blindness. We find this approach interesting since endothelin is implied in the intimal hyperplasia observed in GCA and participates in the vascular occlusion leading to ischemic complications.

We thus added in the conclusion a sentence on possible further works on this topic.

Further works are awaited to better determine and act on the mechanisms implied in cardiovascular events in GCA patients, especially regarding the endothelial dysfunction that leads to the intimal hyperplasia or vascular remodeling. Besides anti-inflammatory or immunosuppressive treatments, future therapeutic approaches developed in atherosclerosis and aiming to limit plaque formation or to reduce the involved inflammatory processes would probably be beneficial in GCA.

Reviewer 2 Report

Overall--I think this review is comprehensive and very well-done.

I have 2 suggestions for consideration:

  1. Although accelerated atherosclerosis is mentioned as a possible contributor to increased CV event rate observed in GCA in the abstract (and there are theoretical reasons why this could occur, as pointed out by the authors) this has not been directly proven in GCA patients, to my knowledge.  Would suggest language be re-framed to acknowledge that the role of accelerated athero (although possible) is uncertain.  One of the main problems, probably, is that athero is generally very common in this age group, so it is difficult to show a difference. Could include the following reference : Gonzalez-Juanatey C, Lopez-Diaz MJ, Martin J, Llorca J, Gonzalez-Gay MA. Atherosclerosis in patients with biopsy-proven giant cell arteritis. Arthritis Rheum. 2007;57(8):1481-1486. doi:10.1002/art.23114, to the section discussing stroke event rate.
  2. In abstract, it is also mentioned that aortic structural changes should be periodically evaluated. In "aorta" section, would be very interesting to include authors' recommendations for how/how often they re-image the aorta in their practice 

Author Response

  1. Although accelerated atherosclerosis is mentioned as a possible contributor to increased CV event rate observed in GCA in the abstract (and there are theoretical reasons why this could occur, as pointed out by the authors) this has not been directly proven in GCA patients, to my knowledge.  Would suggest language be re-framed to acknowledge that the role of accelerated athero (although possible) is uncertain.  One of the main problems, probably, is that athero is generally very common in this age group, so it is difficult to show a difference. Could include the following reference : Gonzalez-Juanatey C, Lopez-Diaz MJ, Martin J, Llorca J, Gonzalez-Gay MA. Atherosclerosis in patients with biopsy-proven giant cell arteritis. Arthritis Rheum. 2007;57(8):1481-1486. doi:10.1002/art.23114, to the section discussing stroke event rate.

Thank you for this constructive comment. We totally agree and rewrote the paragraph concerning mechanisms involved in ischemic events. We also included the reference and stated the uncertainty of accelerated atherosclerosis in GCA.

2.1 Implicated mechanisms

Ischemic events occurring in GCA might be linked to three distinct or intricated mechanisms.

The first mechanism corresponds to the vascular wall inflammation that leads to a narrowing of the vascular lumen and downstream tissue ischemia. The pathophysiology of vascular damage in GCA is to date well understood. Briefly, following a trigger whose origin is still unknown, dendritic cells in the adventitia of arteries are activated and result in the production of chemokines that attract lymphocytes within the vascular wall. A cytokine-mediated activation, proliferation and polarization of T lymphocytes into TH1 and TH17 cells lead to the recruitment of macrophages and T CD8 lymphocytes. In addition, some growth factors such as vascular endothelial growth factor (VEGF) and platelet-derived growth factor (PDGF) increase the proliferation of vascular smooth muscle cells. Altogether, the different recruited and activated cells are responsible for vascular damage and result in an intimal hyperplasia that can progressively obstruct the vascular lumen [19-21].

The second mechanism regards the possible accelerated atherosclerosis in patients with GCA, also documented in systemic lupus erythematosus, rheumatoid arthritis and in some systemic vasculitides [22-25]. Endothelial dysfunction, which is a cornerstone of atherogenesis [26], has been poorly studied in GCA. Measuring the flow-mediated vasodilatation by brachial ultrasonography, which is a marker of endothelial function, two studies provided paradoxical results. One showed an initial impaired flow-mediated vasodilatation that improved with GC [27] supporting the presence of an endothelial dysfunction in GCA patients before treatment. However, these results were not replicated in another study [28].

Another ultrasonography study did not show any increase in the rate of atheromatous carotid plaques during the follow-up of GCA patients when compared to controls [29]. Thus, taken together, no specific study has shown accelerated atherosclerosis in GCA patients, although there is much evidence that systemic inflammation provides a proatherogenic effect by inducing dyslipidemia, insulin resistance, hypercoagulation, endothelial dysfunction and oxidative stress [22]. 

The third mechanism corresponds to the various direct and indirect impacts of GC on atherosclerosis formation. Glucocorticoid excess, as observed in Cushing's syndrome, is associated with an increased risk of CV events [30]. A recent study of 87,794 patients with immune mediated diseases, including 25,581 GCA patients +/- with polymyalgia rheumatica, confirmed that GC use, even with daily doses <5mg of prednisone equivalent, was associated with an increased risk of CV disease [31]. However, the implication of GC in atherogenesis is complex, since the effects of GC were shown to be both athero-protective and pro-atherogenic. On one hand, some preclinical studies showed that GC exposure may reduce the recruitment of monocytes and macrophages within the atherosclerotic lesions and reduce the cholesterol accumulation in macrophages. It also results in a reduction of macrophage-induced pro-inflammatory cytokine release limiting the vascular remodeling [32]. On the other hand, GC are also implied in pro-atherogenic mechanisms. They lead to a direct endothelial cell dysfunction by promoting the production of vasoconstrictors acting on the smooth muscle cells. In addition, some other pre-clinical experimental models suggest an increased accumulation of macrophages in atherosclerotic lesions, with an increased cholesterol esterification [32]. Altogether, long term GC exposure is associated with increased atherosclerosis. However, the exact impact of GC and the mechanisms implied in atherosclerosis formation is more complex, since GC may act at different cellular levels resulting in opposite results, either beneficial or detrimental to the development of atherosclerotic plaques.

  1. In abstract, it is also mentioned that aortic structural changes should be periodically evaluated. In "aorta" section, would be very interesting to include authors' recommendations for how/how often they re-image the aorta in their practice 

This is a key question since different European, French, British or American recommendations do not specify the rhythm and preferred modalities of aorta morphology controls.

Based on the published series, most of the aortic changes occurring during follow-up regard the thoracic section of the aorta. Echocardiography may be a useful, cost-effective and non-irradiating tool to monitor the thoracic aorta caliber and detect dilation. We perform an annual echocardiography in our patients with large-vessel vasculitis, but validation in further studies is required to confirm the usefulness of this practice. In patients with a known aorta dilation, CT angiography is probably also required but the frequency of imaging should probably be adapted to each situation.

In the manuscript, we added at the end of the chapter on aorta complications, page 16:

"Altogether, based on the previous studies showing the frequent development of a dilation mostly on the thoracic section of the aorta in patients with LVV [73-76], a regular monitoring of aorta caliber is required. Echocardiography may be a useful, cost-effective and non-irradiating tool to monitor the thoracic aorta caliber and detect any dilation. An annual echocardiography, associated with the overall assessment of usual CV risk factors in this elderly population, could be suggested. However, given the absence of specific guidelines regarding the best frequency and modality of aorta monitoring, further research is required to confirm the usefulness of this practice in GCA."

Round 2

Reviewer 1 Report

Despite the extensive modifications of the manuscript I feel still confused.

There is no clear categorization in pre-existing CV risk factors steroid induced CV events and vasculitic induced CV-events and truely this may often not be possible in the existing literature. This is not discussed in depth. Maybe we need well controlled cohort studies to answer these questions.  

Author Response

Thank you for your careful new review.

The categorization of the three mechanisms implied in CV events remains hypothetical and conceptual, since the clinician in charge of a GCA patient with a CV event will treat CV risk factors and adapt the anti-inflammatory/immunosuppressive treatment if the vasculitis is active. We do not know how to provide a clearer categorization than the one we did in the paragraph "2.1 implied mechanisms". Moreover, the following paragraph "2.2 Distinction of vascular inflammation and atherosclerosis in ischemic events" discussed some practical points that can help distinguish between inflammation and atherosclerosis.

We totally agree that further works are required to better clarify the respective implication of each mechanism in the CV events occurring in a GCA patient.

This is why we wrote in the conclusion :

"Further works are awaited to better determine and act on the mechanisms involved in cardiovascular events in GCA patients, especially regarding the endothelial dysfunction that leads to the intimal hyperplasia or vascular remodeling. Besides anti-inflammatory or immunosuppressive treatments, future therapeutic approaches developed in atherosclerosis and aiming to limit plaque formation or to reduce the involved inflammatory processes may be beneficial in GCA."
